# Evaluation of Non-Invasive Hemoglobin Monitoring in Perioperative Patients: A Retrospective Study of the Rad-67^TM^ (Masimo)

**DOI:** 10.3390/diagnostics15020128

**Published:** 2025-01-08

**Authors:** Philipp Helmer, Andreas Steinisch, Sebastian Hottenrott, Tobias Schlesinger, Michael Sammeth, Patrick Meybohm, Peter Kranke

**Affiliations:** 1Department of Anaesthesiology, Intensive Care, Emergency and Pain Medicine, University Hospital Würzburg, Oberdürrbacher Str. 6, 97080 Würzburg, Germany; steinisch_a@ukw.de (A.S.); hottenrott_s@ukw.de (S.H.); micha@sammeth.net (M.S.); meybohm_p@ukw.de (P.M.); kranke_p@ukw.de (P.K.); 2Department of Anaesthesiology and Intensive Care, BG Murnau, Professor-Küntscher-Str. 8, 82418 Murnau, Germany; tobias.schlesinger@bgu-murnau.de; 3Department of Applied Sciences and Health, Coburg University of Applied Sciences and Art, Friedrich-Streib-Str. 2, 96450 Coburg, Germany

**Keywords:** co-oximetry, SpHb, point-of-care, anemia, hemoglobin, accuracy, PBM

## Abstract

**Background**: Hemoglobin (Hb) is a crucial parameter in perioperative care due to its essential role for oxygen transport and tissue oxygenation. Accurate Hb monitoring allows for timely interventions to address perioperative anemia and, thus, prevent morbidity and mortality. Traditional Hb measurements rely on invasive blood sampling, which significantly contributes to iatrogenic anemia and poses discomfort and increased infection risks. The advent of non-invasive devices like Masimo’s Rad-67™, which measures Hb using pulse CO-oximetry (SpHb), offers a promising alternative. This study evaluates the accuracy of SpHb compared to clinical standard blood gas analysis (BGA) in perioperative patients. **Methods**: This retrospective study analyzed 335 paired Hb measurements with an interval <15 min between SpHb and BGA in the operating theater and post-anesthesia care unit of a university hospital. Patients experiencing hemodynamic instability, acute bleeding, or critical care were excluded. Statistical analysis included Bland–Altman plots and Pearson correlation coefficients (PCCs) to assess the agreement between SpHb and BGA. Potential confounders, e.g., patient age, skin temperature, sex, perfusion index (PI), and atrial fibrillation, were also analyzed. **Results**: The bias of the SpHb compared to BGA according to Bland–Altman was 0.00 g/dL, with limits of agreement ranging from −2.70 to 2.45 g/dL. A strong correlation was observed (*r* = 0.79). Overall, 57.6% of the paired measurements showed a deviation between the two methods of ≤±1 g/dL; however, this applied to only 33.3% of the anemic patients. Modified Clark’s Error Grid analysis showed 85.4% of values fell within clinically acceptable limits. Sex was found to have a statistically significant, but not clinically relevant, effect on accuracy (*p* = 0.02). **Conclusions**: The Rad-67^TM^ demonstrates reasonable accuracy for non-invasive SpHb, but exhibits significant discrepancies in anemic patients with overestimating low values. While it offers potential for reducing iatrogenic blood loss, SpHb so far should not replace BGA in critical clinical decision-making.

## 1. Introduction

Hemoglobin plays a crucial role in oxygen transport and tissue oxygenation, making it a critical parameter in perioperative care [1]. Anemia, characterized by decreased hemoglobin levels, can impair tissue oxygenation and increase mortality, particularly in vulnerable perioperative patients [2]. These individuals face an increased risk of bleeding, which often leads to anemia and significantly worse patient outcomes [3]. Hemorrhagic events are, thus, a major contributor to perioperative mortality [4,5]. Consequently, the close monitoring of hemoglobin levels in perioperative settings is essential for timely interventions and optimizing patient safety.

Traditionally, hemoglobin is measured by invasive blood sampling methods for point-of-care diagnostics such as blood gas analysis (BGA). These methods require a blood sample to be drawn from the patient, which either requires invasive catheters or venous puncture associated with discomfort and increased infection risk [6,7]. Frequent invasive blood draws contribute significantly to iatrogenic blood loss and can exacerbate anemia, counteracting the principles of Patient Blood Management (PBM) [8]. Thereby, PBM is a patient-centered interdisciplinary approach that focuses on patient safety and improving patient outcomes [9]. It is based on three key pillars with more than 100 specific aspects focusing on (i) the management of anemia with optimizing coagulation, (ii) the minimization of iatrogenic and unnecessary blood loss, and (iii) patient-centered decision-making, especially with the use of blood transfusions [10]. Frequent invasive blood draws contribute significantly to iatrogenic blood loss and can exacerbate anemia, counteracting the principles of Patient Blood Management (PBM).

The development of non-invasive hemoglobin measurement devices, such as Masimo’s Rad-67™, offers a promising solution to reduce the need for invasive blood draws and, therefore, minimize iatrogenic blood loss. The device utilizes pulse CO-oximetry, a continuous, non-invasive method to measure total hemoglobin concentration (SpHb) in the blood. The Rad-67™ sensor, placed on the patient’s finger, emits various wavelengths of light through the tissue and detects changes in light absorption, allowing for real-time hemoglobin monitoring without the need for invasive blood drawings.

Non-invasive hemoglobin monitoring offers several advantages. Continuous SpHb monitoring is associated with reduced intraoperative transfusions, lower postoperative bleeding rates, and shorter ICU stays [11]. In a recent study, the percentage of patients needing a transfusion decreased by 7.4% when Hb was continuously monitored, and the number of transfused units per patient decreased by 12.6% [12]. Furthermore, pediatric patients undergoing cranioplasty operations with continuous SpHb monitoring experienced fewer transfusions and better postoperative outcomes [13,14]. Similarly, the implementation of SpHb monitoring has even reduced mortality in various surgical settings, including cesarean sections and neurosurgery [15,16]. Furthermore, SpHb monitoring has been shown to decrease the incidence of unnoticed anemia, adhere to an Hb target range, and reduce inappropriate blood transfusions, contributing to better patient care and reduced health care costs [17,18,19].

Given the potential benefits of non-invasive hemoglobin monitoring in specific areas of care, we conducted a retrospective study to evaluate the measurement accuracy of the Rad-67™ device in general in perioperative patients, as the Rad-67™ was routinely used in our department. Our primary outcome was to investigate the accuracy of SpHb measurements compared to the clinical standard BGA and, secondarily, to identify factors influencing the device’s performance in a real-world clinical setting.

## 2. Methods

### 2.1. Study Design

This retrospective study was conducted in the operating theater and post-anesthesia care unit at the Department of Anaesthesiology, Intensive Care, Emergency, and Pain Medicine at the University Hospital Würzburg, Germany. Data collection and analysis were performed in accordance with Article §27 of Bavarian hospital law and the local ethics committee of the University of Würzburg approved the analysis (ref. no. 20240216-01). All simultaneous hemoglobin (Hb) measurements with an interval of less than 15 min, using both the Rad-67™ Pulse CO-Oximeter^®^ and the clinical standard BGA, GEM Premier 5000 (Instrumentation Laboratory Company, Bedford, MA, USA), were included in the analysis. Measurements from patients who were hemodynamically unstable, suffering from acute bleeding, or receiving critical care were excluded.

Both measurement techniques were employed as part of the clinical standard between 2021 and 2023 in our department. The primary objective of this study was to investigate the measurement accuracy of the non-invasive SpHb measurements obtained by the Rad-67™ compared to the invasive clinical standard, defined as BGA. The secondary objective was to investigate factors potentially affecting measurement accuracy. The study was designed, conducted, and analyzed without financial support or any other involvement from industrial partners in order to prevent conflicts of interest. All employed devices, including the Rad-67™ device and all sensors, were purchased through regular channels.

### 2.2. Measurements of the Hemoglobin

In this study, we compared the measurement accuracy of a non-invasive spot-method for hemoglobin (SpHb) with the established clinical standard of point-of-care invasive hemoglobin (Hb) measurement employing blood gas analysis (BGA). The non-invasive SpHb was measured using the Rad-67™ Pulse CO-Oximeter^®^ (Masimo, Irvine, CA, USA), a certified Class B medical device (IEC 60601-1-2:2007) authorized according to ANSI/AAMI ES 60601-1:2005 and EN/ISO 80601-2-61:201. A Rainbow^®^ Super DCI^®^-mini sensor was placed on the patient’s index finger, utilizing (i) spectrophotometry, (ii) photoplethysmography (PPG), and (iii) Signal Extraction Technology (SET^®^). The Rad-67™ device uses multiple light-emitting diodes (LEDs) that emit various wavelengths of visible and infrared light (500–1400 nm), which pass through the tissue to a detector diode on the opposite site [20].

(i)Spectrophotometry is a method used to measure how much light is absorbed by different hemoglobin species within the blood. The principle is based on the fact that various hemoglobin types (oxyhemoglobin, deoxyhemoglobin, carboxyhemoglobin, methemoglobin) and other blood plasma components absorb light at distinct wavelengths. By passing visible and infrared light through the tissue and measuring the light absorption, the Rad-67™ distinguishes between these hemoglobin types. Each compound has a unique absorption spectrum, and the detector captures the amount of light that passes through the blood. The difference in light absorption at specific wavelengths allows for the identification and quantification of the various hemoglobin types.(ii)Photoplethysmography (PPG) is an optical technique that detects changes in blood volume in the tissue over time, based on light absorption related to the pulsatile nature of blood flow [21]. The PPG signal consists of a pulsatile component (‘AC’) due to cardiac activity and a slowly varying baseline (‘DC’) influenced by factors such as respiration and thermoregulation. By detecting the changes in light absorption caused by variations in arterial blood volume, the device can distinguish between arterial and venous blood.(iii)The signal extraction technique (SET^®^) is employed to separate the arterial signal from noise and interference, such as motion artifacts [22]. By improving the signal-to-noise ratio, SET^®^ assists the device in providing more reliable measurements of SpHb.

This combination of (i) to (iii) and the use of parallel algorithms are employed in the Rad-67™ to analyze the raw data for outputting the SpHb.

For the invasive clinical standard point-of-care method, approximately 1 mL of arterial or venous blood was drawn via a placed 20G arterial line (Arrow, Teleflex Medical, Wayne, PA, USA or Insyte-W, BD Medical, Franklin Lakes, NJ, USA) or a 22-16G venous catheter (Vasofix Safety Cannulae, B Braun). The blood samples were analyzed using the GEM 5000 Premier (Instrumentation Laboratory Company, Werfen, Bedford, MA, USA), which uses a spectrophotometric method that operates over a broader range of wavelengths compared to the non-invasive device, making it less susceptible to interference and yielding a direct measurement of hemoglobin without the need for motion artifact correction or pulsatile flow differentiation.

### 2.3. Statistical Analysis

All routine data were extracted from the patient data management system (PDMS) “Copra” (COPRA System GmbH, Berlin, Germany, V6.84.20) and exported to an Excel file. The primary endpoint of this study was assessed using Bland–Altman plots [23] and linear regression analyses, employing the Pearson Correlation Coefficient (PCC). For secondary endpoints, scatter plots were used for continuous variables, and box plots were used for categorical variables. For the evaluation of possible effects of the investigated attributes and factors, linear regression analysis was applied to continuous variables, while differences in the categorical variables were evaluated using the Kolmogorov–Smirnov test.

Descriptive statistics were computed as both the mean with the corresponding standard deviation (SD) and the median with the interquartile range (IQR). All statistical analyses were performed using the R platform (v4.2.0). Bland–Altman plots were generated to assess the presence of systematic biases, with the values obtained by the reference method (BGA) subtracted from the corresponding Rad-67™ hemoglobin readings. Linear regression analysis was performed to evaluate the correlation between the two measurement methods, employing PCC.

For the enhanced interpretability of the results, we also applied a modified version of Clark’s error grid, originally developed to compare blood glucose monitoring devices [24]. The advantage of Clark’s error grid lies in its ability to present not only numerical deviation, but also the clinical relevance of the discrepancy. The grid categorizes differences between the test and reference measurements into three zones: green for clinically insignificant differences, yellow for potential clinical relevance, and red for significant clinical risk with potential patient harm. Notably, Mills et al. had already utilized this method for comparing different Hb measurement techniques [25]. To further account for Hb thresholds relevant to patient blood management (PBM), we self-designed a modified version of the Clark’s error grid with modified boundaries of these zones. Particular emphasis was placed on diagnostic thresholds for mild anemia, as these patients should be monitored perioperatively for factors such as iron deficiency. The early identification and timely treatment of modifiable factors, like iron deficiency, can improve postoperative recovery. Furthermore, we broaden the red zones, based on the likelihood of causing patient harm, because in our opinion, e.g., a measurement of Hb 7.5 g/dL by the clinical standard and a reading of 4.5 g/dL by a test device results in a relevant clinical risk with potential patient harm.

For data visualization, we employed the ggplot2 package (v3.3.6) to generate box plots and scatter plots. Descriptive statistics, including the arithmetic mean, median, quartiles, and IQR, were calculated using the R function summary(). The Pearson correlation and linear regression analyses were performed using cor.test() and lm(), respectively. Bland–Altman scatter plots were designed to plot the mean of each measurement pair against the real error, with bias estimated as the average of these errors. Limits of agreement (LoAs) were calculated by applying an offset of twice the corrected sample standard deviation to the mean errors. Confidence intervals for bias and LoAs were calculated assuming a Student’s t-distribution using the R function qt().

## 3. Results

### 3.1. Overview of the Cohort

A total of 335 simultaneous hemoglobin measurements were included in our analysis. The cohort consisted of 59.4% male and 40.6% female patients, with a median age of 65.3 years (range: 19 to 98 y.o.). Of our cohort, 39 patients (11.6%) presented with severe anemia (hemoglobin ≤ 9 g/dL), while 177 (52.8%) showed mild anemia (hemoglobin < 13 g/dL) [26], and 119 (35.5%) had no anemia (hemoglobin ≥ 13 g/dL). The median perfusion index (PI) was 4.1 (IQR: [1.6; 7.4]), and the median peripheral oxygen saturation (SpO_2_) was 97.0% (IQR: [94; 98]), ranging from a minimum of 79% to a maximum of 100%. The majority of patients had sinus rhythm (*n* = 303), while 22 patients were diagnosed with atrial fibrillation, and 10 patients with unknown heart rhythm. Hypothermia, defined as a body temperature < 36 °C, was observed in 19 patients, and the temperature was unrecorded in 14 patients. No patients suffered from hyperthermia, defined as a body temperature > 38.5 °C. Measurements were collected both intraoperatively (*n* = 54) and in the post-anesthesia care unit (*n* = 281).

### 3.2. Measurement Accuracy

We evaluated the measurement accuracy of the RAD-67™ in comparison to the clinical standard blood gas analysis (BGA) using Bland–Altman analysis and a linear regression model. Among the 335 paired measurements, the difference of the hemoglobin readings was greater than +1 g/dL in 75 pairs, less than or equal to ±1 g/dL in 193 pairs, and greater than −1 g/dL in 67 pairs. Therefore, 57.6% of all analyzed measurements showed a good clinical accuracy with a difference less than ±1 g/dL.

The bias calculated from the Bland–Altman plot was 0.00 g/dL (95% CI: −0.14 to 0.15 g/dL), with limits of agreement (LoAs) ranging from −2.70 g/dL (95%CI: −2.95 to −2.44 g/dL) to 2.45 g/dL (95%CI: 2.45 to 2.95 g/dL) (Figure 1). The correlation between RAD-67™ and BGA was strong, with a Pearson correlation coefficient of *r* = 0.79. The linear regression analysis yielded a slope of 0.63 and a shift of 4.35 g/dL (Figure 2).

However, when focusing on patients suffering from severe anemia defined as Hb ≤ 9 g/dL (*n* = 42), only 33.3% of the paired measurements showed a good clinical accuracy with a difference less than ±1 g/dL. The bias in the anemic cohort was −1.2 g/dL (95% CI: −1.65 to −1.04 g/dL).

To assess the clinical relevance of the observed differences, we employed a modified Clark’s Error Grid, adjusting for hemoglobin thresholds as recommended by patient blood management (PBM) guidelines (Figure 3). Based on this modified grid, 85.4% of the paired measurements fell into the green category (results within a clinically acceptable range), 14.6% into the yellow category (results within a clinically significant error), and 0% into the red category (results outside a clinically acceptable range). When applying the standard Clark’s Error Grid, 92.2% of the values were classified as green and 7.8% as yellow.

### 3.3. Potential Confounders

Furthermore, we investigated the influence of different attributes, including patient age, perfusion index (PI), skin temperature, sex, and atrial fibrillation on the measurement accuracy of the RAD-67™ compared to the clinical standard (Figure 4 and Figure 5). None of the analyzed attributes had a clinically significant impact on the measurement accuracy. However, we observed a statistically significant impact of sex (*p* = 0.02). The mean absolute error was 0.3 g/dL in men and −0.1 g/dL in women.

## 4. Discussion

We investigated the measurement accuracy of the Masimo RAD-67™ for non-invasive hemoglobin (SpHb) monitoring in comparison to the established clinical standard of invasive blood gas analysis (BGA) in a perioperative setting. Our cohort comprised 335 consecutive Hb measurements. While our findings indicate that the RAD-67™ device demonstrates reasonable overall accuracy, clinical caution is warranted due to inaccuracies in patients with severe anemia.

The correlation coefficient between SpHb and BGA was *r* = 0.79, with a bias of 0.0 g/dL and limits of agreement (LoAs) ranging from −2.70 g/dL to 2.45 g/dL. Although this suggests a reasonable level of agreement, it is crucial to note that low hemoglobin values tended to be overestimated, while high values were underestimated by the RAD-67™. This systematic error, reflected by a slope of 0.63 and a shift of 4.35 in the linear regression model, represents a clinical risk, particularly as overestimation of low hemoglobin levels may lead to delayed or insufficient therapy in patients at risk of anemia-related complications. Consequently, SpHb measurement should not be used in patients with high perioperative blood loss or known severe anemia. Additionally, the use of SpHb in patients at high risk of anemia should be carefully considered for critically ill patients, given the high prevalence of anemia in intensive care settings and the potential risks of Hb overestimation.

Furthermore, we observed that none of the investigated attributes had a clinically relevant impact on the measurement accuracy of the RAD-67™. While sex showed a statistical significant effect (*p* = 0.02), as previously described in another study, we declared the impact as clinically irrelevant [27]. The PI, representing the ratio of pulsatile to non-pulsatile blood flow, has been previously described as a factor influencing measurement accuracy [28]. However, in our cohort, variations in PI did not affect the performance of the device. Similarly, no significant differences were observed in patients with atrial fibrillation, despite the manufacturer’s advice not to use the device in patients with cardiac arrhythmias. Although patient age does not significantly impact measurement accuracy, particular caution is advised when using the device in elderly patients due to their higher prevalence of anemia, a condition in which the device demonstrates reduced performance.

Comparison with previous studies reveals a heterogeneous body of evidence (Table 1) [14,16,18,20,25,27,29,30,31,32,33,34,35,36,37]. Studies evaluating SpHb measurements with Masimo devices in different patient cohorts and clinical settings have reported bias ranging from 0.0 g/dL (intraoperative) to 2.2 g/dL (obstetric anesthesia), and correlations between *r* = 0.55 (children aged 1–5 years) and *r* = 0.95 (intraoperative) [20,25,30,34]. Further contextualizing our results, 7.8% of our data points fell within the “yellow zone” of the standard Clark’s error grid, with no measurements falling into the “red zone,” indicating a better performance compared to previous reports in obstetric anesthesia with up to 15.6% of data points in the yellow zone [25]. Comparing our proportions of the absolute error with a previous study reveals that 57.6% vs. 68.6% of the measurements showed a difference ≤±1 g/dL between SpHb and BGA Hb [27]. However, they reported that SpHb was over-reported with >1 g/dL in 18.6% vs. 22.4% of the measurements in our analysis. Furthermore, it is essential to acknowledge that even Hb measurements obtained by BGA are not entirely free from bias when compared to the laboratory gold standard, with one study reporting a BGA bias of −0.38 g/dL [38].

The manufacturer explicitly states that SpHb monitoring should not replace laboratory-based hemoglobin measurements and that clinical decisions should be based on confirmation by gold standard blood sample analysis. Our findings reinforce this guidance, as 40.9% of the SpHb readings in our study fell outside the manufacturer’s stated accuracy range of ±1 g/dL for hemoglobin levels between 8 and 17 g/dL. These deviations are clinically significant and could impact decision-making, especially in cases wherein prompt and accurate hemoglobin monitoring is crucial.

### Limitations

Our study has several limitations, especially due to its retrospective design. First, we cannot draw any conclusions about the device performance in repeated measurements, as only one measurement per patient was available. As a result, no conclusions can be drawn regarding the reproducibility of measurements within individual patients. Additionally, our patient data management system (PDMS) only documented successful measurements, so we have no information on failed measurement attempts, including the number of attempts or their reasons for failure. Furthermore, we exclusively included stable patients who were not receiving vasoactive drugs, preventing us from drawing conclusions about critically ill patients. The cohort is also sex-unbalanced, with more men than women, and no structured data regarding skin color was available. This limitation precludes any assessment of potential influences of skin pigmentation on the measurement accuracy.

Another limitation concerns the unknown measurement side in our cohort. According to the manufacturer, certain factors, such as wearing a blood pressure cuff on the same arm, can affect accuracy, but we were unable to evaluate this, as the measurement side was not documented. Finally, the manufacturer indicates that the device may be inaccurate in patients with tricuspid regurgitation, a disease we cannot evaluate in our cohort due to the lack of systematic documentation of pre-existing diseases in our PDMS.

## 5. Conclusions

In conclusion, while the RAD-67™ offers a non-invasive alternative for hemoglobin monitoring, its accuracy is variable, especially at low hemoglobin values. Clinical use should be accompanied by confirmatory testing using established methods, particularly when hemoglobin levels are critical for treatment decisions. However, the RAD-67™ serves as a valuable supplementary tool for orientation in patients with unknown hemoglobin values, such as during urgent operations without recent laboratory results. Future research should continue to explore the utility of non-invasive hemoglobin monitoring in specific patient populations and refine the technology to improve its clinical reliability.

## Figures and Tables

**Figure 1 diagnostics-15-00128-f001:**
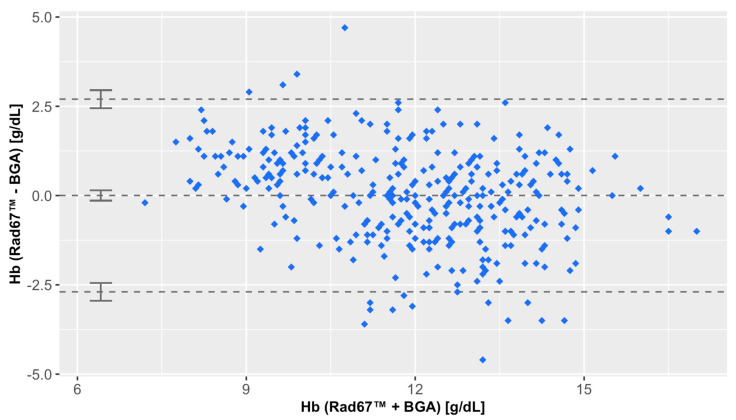
Bland–Altman plot comparing Hb measurements by the Rad-67™ (Masimo) with the clinical standard method (BGA). Scatterplots show the real errors of the measurements (y axis: Rad-67™ measurements minus BGA reference) stratified by the mean of each measurement pair (x axis). Dashed horizontal lines mark the bias, i.e., the arithmetic average of all real errors with the limits of agreement (LoAs) as determined by an offset of ±2 times the standard deviation (SD). Error bars show the 95% confidence interval (CI) for the bias and both LoAs.

**Figure 2 diagnostics-15-00128-f002:**
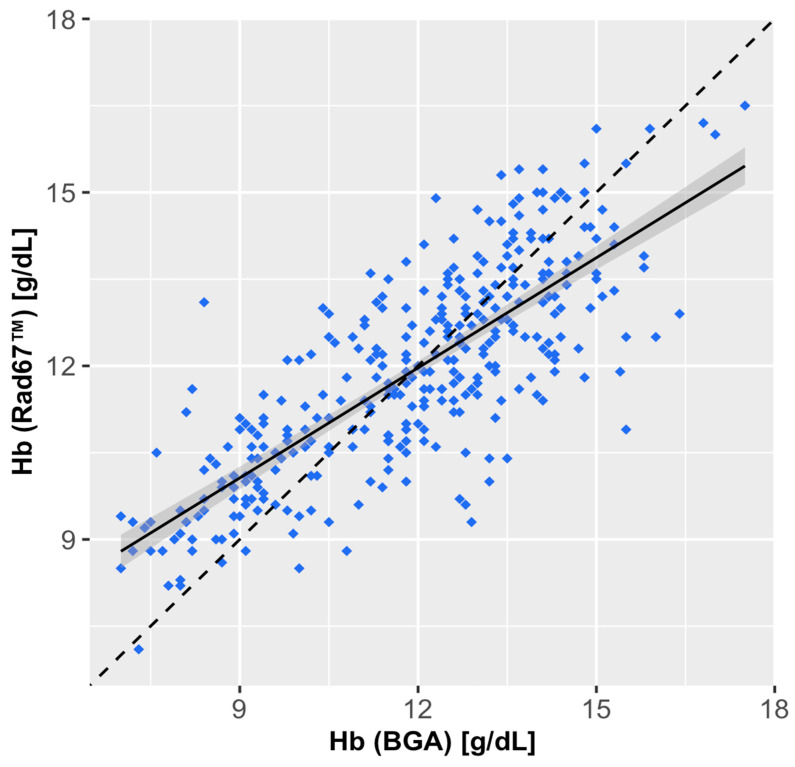
Linear correlation assessment of the Hb measurements comparing the Rad-67™ to BGA. Scatterplots localize each of the paired measurements by the Hb reference BGA (x axis) and the corresponding Hb measurement of the benchmarked device (y axis). The black solid line depicts the linear regression model, with the 95% confidence interval shaded in gray. The black dotted line depicts the perfect regression.

**Figure 3 diagnostics-15-00128-f003:**
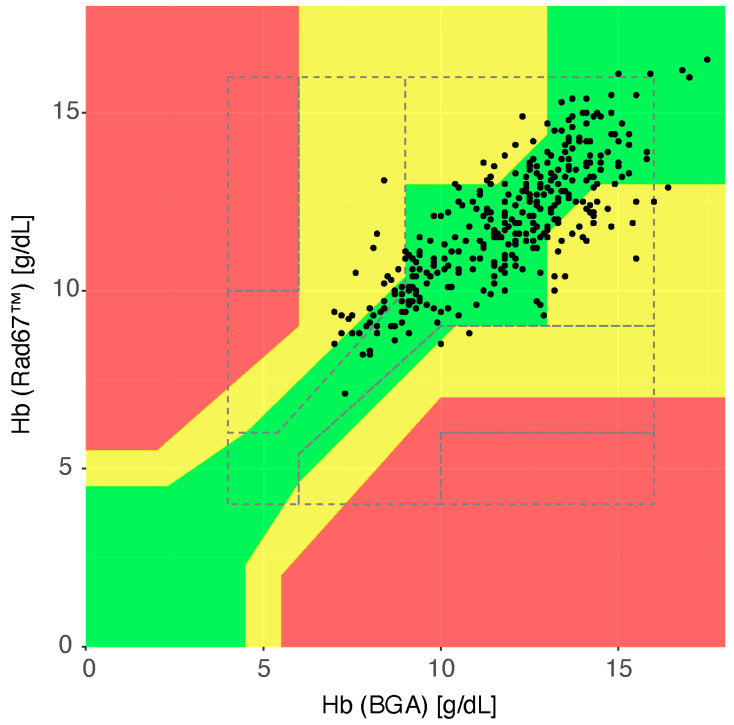
Self-modified Clark’s error grid to quantify clinical accuracy of the Hb measurements. The presented plot is adapted to Hb threshold values of the PBM. The dashed lines show the original Clark’s error grid. On the x axis of the scatter plot, the absolute Hb values of the reference method are shown, and on the y axis, the investigated device. Values within the green zone are considered clinically acceptable, measurement pairs in the yellow zone indicate a clinically relevant deviation, and those in the red zone correspond to severe clinically relevant deviations.

**Figure 4 diagnostics-15-00128-f004:**
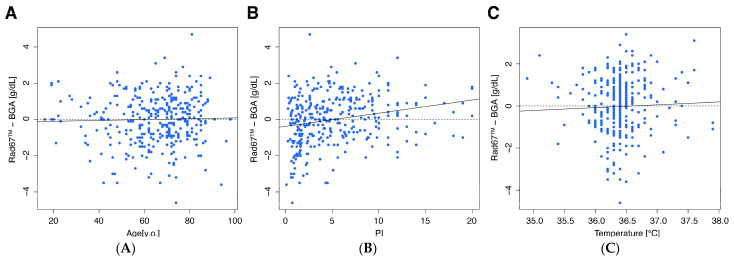
Scatterplots showing the absolute difference between the reference method subtracted from the investigated device (y axis) in the context of different continuous variables (x axis) to assess potential influences. (**A**): age [y.o.], (**B**): PI (Perfusion Index), and (**C**): temperature [°C]. The linear correlation is shown as a solid line, and the baseline is shown as a dotted line.

**Figure 5 diagnostics-15-00128-f005:**
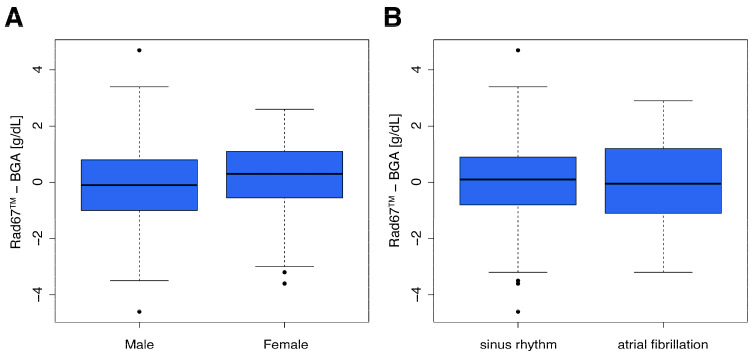
Box plot visualizations of the absolute errors binned by categorical classifications of the patient attributes. (**A**): gender; (**B**): atrial fibrillation.

**Table 1 diagnostics-15-00128-t001:** Overview of studies investigating the measurement accuracy of Co-Oximetry by Masimo for SpHb. The first author, the year of publication, the employed device, the number of analyzed patients, the description of the study cohort, the bias with the corresponding limits of agreement (if available), and the correlation are given.

Author	Year	Device	Number	Cohort	Bias	Correlation
Hornedo-González, K.D. [29]	2023	Masimo Pronto Pulse Co-Oximeter	112	preoperative elective surgical patients	0.8 g/dL (LoA −2.2; 3.9 g/dL)	NA
Al Aseri, Z.A.A. [31]	2023	Masimo Rad-67™	650	emergency department	0.146 g/dL (LoA −2.58; 2.87 g/dL)	*r* = 0.812
Arai, Y. [20]	2023	Masimo Rad-67™	102	early childhood	0.188 (LoA −1.61; 1.99 g/dL)	*r* = 0.548
Caulfield, KC. [37]	2023	Masimo Rad-67™	180	neurosurgery patients	−1.4 g/dL (LoA −3.59; 0.79 g/dL)	NA
Mills, K. [25]	2023	Masimo Rad-67™	301	pregnant people on the day of childbirth	2.2 g/dL (LoA 0.08; 4.30 g/dL)	ICC 0.4
Beleta, M.I. [30]	2022	Masimo pulse co-oximetry	60	elective CS under general anesthesia	0.348 g/dL	*r* = 0.946
Bıcılıoğlu, Y. [32]	2022	-	110	patients aged 1–5 years	0.3 g/dL	*r* = 0.675
Ke, Y.H. [27]	2021	Masimo Rad-67™	392	screening pre-operative anaemia	0.14 g/dL (LoA 2.24; −1.95 g/dL)	*r* = 0.76
Tang, B. [14]	2019	Masimo Radical-7™	28	children with thalassemia	−0.29 g/dL (LoA −2.30; 1.72 g/dL)	*r* = 0.69
Welker, E. [33]	2018	Masimo Radical-7™	21	spine surgery on multiple spinal segments and cytoreductive surgery	0.80 g/dL (LoA 3.94; −2.33)	NA
Adel, A. [34]	2018	Masimo Radical-7™	210	pediatric trauma patients	0.01 g/dL (LoA −1.33; 1.34 g/dL)	*r* = 0.938
Gamal, M. [35]	2018	Masimo Radical-7™	184	primary or revision total hip or total knee arthroplasty	0.12 g/dL (LoA −0.56; 0.79 g/dL)	*r* = 0.872
García-Soler, P. [36]	2017	Masimo Radical-7™	284	trauma patients with low hemoglobin levels	0.07 g/dL (LoA −2.26; 3.59 g/dL)	*r* = 0.72
Martin, J. [18]	2016	Masimo Pronto Pulse Co-Oximeter	100	general anesthesia with complete muscle relaxation, adult	−0,2 g/dL (LoA −3.4; 4.4 g/dL)	CCC = 0.69
Awada, W.N. [16]	2015	Masimo Radical-7™	83	pediatric Intensive Care Unit	0.0 g/dL (LoA −1.6; 1.5 g/dL)	NA

## Data Availability

The data supporting the findings of this study are available from the corresponding author upon reasonable request and for research purposes only. Data will not be shared for commercial use.

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
