# Peer review of "Evaluation of Non-Invasive Hemoglobin Monitoring in Perioperative Patients: A Retrospective Study of the Rad-67TM (Masimo)"

_diagnostics, 2025, doi:10.3390/diagnostics15020128_

Round 1
Reviewer 1 Report
Comments and Suggestions for Authors
The manuscript presents a timely and relevant investigation into the accuracy of non-invasive hemoglobin (Hb) monitoring using Masimo's Rad-67™ compared to traditional blood gas analysis (BGA) in perioperative patients. The study addresses a critical issue in perioperative care, namely the prevention of iatrogenic anemia and its associated complications.
v The methodology is robust, utilizing a substantial sample size of 335 paired measurements, and the statistical analyses, including Bland-Altman plots and Pearson correlation coefficients, are appropriate for assessing the agreement between the two measurement techniques. The exclusion criteria are well-defined, ensuring that the results are applicable to a stable patient population, which enhances the validity of the findings.
v The results indicate a reasonable level of accuracy for SpHb, with a bias of 0.00 g/dL and a strong correlation (r=0.79) with BGA. However, the significant discrepancies observed in anemic patients, where only 33.3% of these patients had measurements within ±1 g/dL, raise important concerns regarding the reliability of SpHb in this specific population. This limitation should be emphasized in the discussion, as it has direct implications for clinical practice, particularly in settings where timely and accurate Hb monitoring is critical.
v The conclusion appropriately highlights the potential of SpHb to reduce iatrogenic blood loss, but it rightly cautions against its replacement of BGA in critical decision-making scenarios. It would be beneficial for the authors to elaborate on the clinical implications of their findings, particularly how they envision integrating SpHb into perioperative protocols while maintaining patient safety.
v Overall, the manuscript is well-structured and provides valuable insights into the evolving landscape of Hb monitoring. However, further discussion on the limitations of SpHb in specific patient populations, particularly those with anemia, would strengthen the manuscript and provide clearer guidance for clinicians considering the use of non-invasive monitoring technologies.
Author Response
Comments by Reviewer 1:
The manuscript presents a timely and relevant investigation into the accuracy of non-invasive hemoglobin (Hb) monitoring using Masimo's Rad-67™ compared to traditional blood gas analysis (BGA) in perioperative patients. The study addresses a critical issue in perioperative care, namely the prevention of iatrogenic anemia and its associated complications.
v The methodology is robust, utilizing a substantial sample size of 335 paired measurements, and the statistical analyses, including Bland-Altman plots and Pearson correlation coefficients, are appropriate for assessing the agreement between the two measurement techniques. The exclusion criteria are well-defined, ensuring that the results are applicable to a stable patient population, which enhances the validity of the findings.
v The results indicate a reasonable level of accuracy for SpHb, with a bias of 0.00 g/dL and a strong correlation (r=0.79) with BGA. However, the significant discrepancies observed in anemic patients, where only 33.3% of these patients had measurements within ±1 g/dL, raise important concerns regarding the reliability of SpHb in this specific population. This limitation should be emphasized in the discussion, as it has direct implications for clinical practice, particularly in settings where timely and accurate Hb monitoring is critical.
v The conclusion appropriately highlights the potential of SpHb to reduce iatrogenic blood loss, but it rightly cautions against its replacement of BGA in critical decision-making scenarios. It would be beneficial for the authors to elaborate on the clinical implications of their findings, particularly how they envision integrating SpHb into perioperative protocols while maintaining patient safety.
v Overall, the manuscript is well-structured and provides valuable insights into the evolving landscape of Hb monitoring. However, further discussion on the limitations of SpHb in specific patient populations, particularly those with anemia, would strengthen the manuscript and provide clearer guidance for clinicians considering the use of non-invasive monitoring technologies.
Letter to the Reviewer 1:
- Thank you for recognizing the relevance and importance of our investigation into the accuracy of non-invasive hemoglobin (SpHb) monitoring using Masimo's Rad-67™ compared to traditional blood gas analysis (BGA). We appreciate your acknowledgment.
- We are grateful for your positive feedback on the robustness of our methodology.
- We agree with your observation regarding the limitations of SpHb in anemic patients. To address this, we have revised the discussion to emphasize this limitation more explicitly. Additionally, we have clarified that SpHb measurement should not be used in patients with high perioperative blood loss or known anemia. We have also highlighted the clinical risks associated with overestimating low hemoglobin values, which could delay or compromise treatment decisions in these patients (please see Discussion p. 10, line 267 to 271 :
“Consequently, SpHb measurement should not be used in patients with high perioperative blood loss or known severe anemia. Additionally, the use of SpHb in patients at high risk of anemia should be carefully considered or critically ill patients, given the prevalence of anemia in intensive care settings and the potential risks of Hb overestimation.”
- We appreciate your suggestion to elaborate on the clinical implications of our findings. In response, we have revised the conclusion to clarify the role of SpHb in perioperative protocols. Specifically, we have noted that while the RAD-67™ should not replace confirmatory testing for critical decision-making, it can serve as a valuable supplementary tool for “orientation” in patients with unknown hemoglobin levels, such as during urgent operations without recent laboratory testing (please see Conclusion, p. 12 Line 337 to 339):
“However, the RAD-67™ serves as a valuable supplementary tool for orientation in patients with unknown hemoglobin values, such as during urgent operations without recent laboratory results.”
- Thank you for your suggestion to deepen the discussion of SpHb's limitations in specific patient populations. We have revised the manuscript to provide clearer guidance for clinicians, emphasizing the need for careful consideration of SpHb use in critically ill patients and those at high risk of anemia. This includes a statement on the importance of confirmatory testing in these settings to ensure patient safety. The corresponding changes to the discussion can be seen under Answer 3. Furthermore, we added the following statement about the use in the eldery population (please see Discussion, p. 10 line 281 to 283):
“Although patient age does not significantly impact measurement accuracy, particular caution is advised when using the device in elderly patients due to their higher prevalence of anemia, a condition in which the device demonstrates reduced performance.”
Reviewer 2 Report
Comments and Suggestions for Authors
This paper evaluates non-invasive hemoglobin monitoring in perioperative patients using the Masimo Rad-67 device, comparing its accuracy to clinical standard blood gas analysis and identifying factors influencing its performance. The paper is very well-written and technically sound. However, it could be further improved by visualizing hemoglobin trends for individual patients as the function of time together with reference measurements, analyzing them, and discussing the underlying reasons for proportional bias observed in the results.
Comments:
1. Many confounding factors were taken into account. However, not all. One of the significant confounding factors in the case of photoplethysmography sensors is skin color. The authors acknowledge that not considering skin color among the confounding factors is a study limitation. Could skin color potentially influence non-invasive hemoglobin monitoring?
2. A proportional bias can be observed in Figure 1. Is it possible to correct such a bias? Please discuss.
3. The fonts of axes, labels, and ticks are too small in Figures 4, 5.
Author Response
Comments of Reviewer 2:
This paper evaluates non-invasive hemoglobin monitoring in perioperative patients using the Masimo Rad-67 device, comparing its accuracy to clinical standard blood gas analysis and identifying factors influencing its performance. The paper is very well-written and technically sound. However, it could be further improved by visualizing hemoglobin trends for individual patients as the function of time together with reference measurements, analyzing them, and discussing the underlying reasons for proportional bias observed in the results.
Comments:
- Many confounding factors were taken into account. However, not all. One of the significant confounding factors in the case of photoplethysmography sensors is skin color. The authors acknowledge that not considering skin color among the confounding factors is a study limitation. Could skin color potentially influence non-invasive hemoglobin monitoring?
- A proportional bias can be observed in Figure 1. Is it possible to correct such a bias? Please discuss.
- The fonts of axes, labels, and ticks are too small in Figures 4, 5.
Letter to the Reviewer 2:
Introduction:
Thank you for your positive review of our manuscript and your insightful suggestion regarding the visualization and analysis of hemoglobin trends over time for individual patients. We agree that such an analysis would provide valuable insights into the longitudinal performance of the device and the underlying reasons for proportional bias. However, as correctly noted in the methods and limitations section of our manuscript, only one measurement per patient was available in our study cohort. Consequently, we were unable to analyze repeated measurements or assess reproducibility within individual patients. We appreciate your suggestion and acknowledge the importance of longitudinal data in future studies to better understand device performance over time and in varied clinical scenarios. We have already addressed this limitation in the manuscript with the following statement: "First, we can not draw any conclusion about the device performance in repeated measurements, as only one measurement per patient was available. As a result, no conclusions can be drawn regarding the reproducibility of measurements within individual patients."
- Thank you for raising the question of skin color as a potential confounding factor in non-invasive hemoglobin measurement. While it is known that skin color can influence the accuracy of photoplethysmography-based measurements such as SpOâ‚‚ (Sjoding et al., 2020), evidence regarding its impact on non-invasive hemoglobin measurement is heterogeneous and inconclusive.
For example, a study in pediatric patients reported an effect of darker skin color on measurement accuracy, defined as a decrease of -0.13 of the regression coefficient but not statistically significant p=0.3 (Phillips et al., 2015). Another study suggested that non-invasive hemoglobin monitoring remains a promising tool even in dark-skinned critically ill patients with low hemoglobin levels (Murphy et al., 2018). Furthermore, Masimo claims that skin pigmentation does not affect the accuracy of their devices (https://www.masimo.com/technology/pulse-oximetry/skin-pigmentation/). Although we already considered to further discuss the effect of skin color in our manuscript, but we decided against it because of the lack of evidence given in our study. We believe an in-depth discussion of skin color's influence is beyond the scope of this manuscript, as we cannot draw any conclusions based on our dataset. Nonetheless, we have stated in the manuscript that this is a limitation in our study.
- We agree that proportional bias is observed in the results, with low hemoglobin levels being overestimated and high values underestimated, as shown in Figure 2. However, we believe a general correction for this bias is not feasible due to the high variability in the measurements. As observed, no systematic bias is apparent in Figure 1, and the scatter in the data precludes the application of a reliable correction method.
- Thank you for pointing out that the fonts for axes, labels, and ticks are too small in Figures 4 and 5. We will adjust these elements to improve readability and ensure clarity in all figures during the final editing, while uploading the vector graphics.
References (alphabetic):
- Murphy SM, Omar S. The Clinical Utility of Noninvasive Pulse Co-oximetry Hemoglobin Measurements in Dark-Skinned Critically Ill Patients. Anesth Analg. 2018 May;126(5):1519-1526. doi: 10.1213/ANE.0000000000002721. PMID: 29239951
- Phillips MR, Khoury AL, Bortsov AV, Marzinsky A, Short KA, Cairns BA, Charles AG, Joyner BL Jr, McLean SE. A noninvasive hemoglobin monitor in the pediatric intensive care unit. J Surg Res. 2015 May 1;195(1):257-62. doi: 10.1016/j.jss.2014.12.051. Epub 2015 Jan 9. PMID: 25724765; PMCID: PMC5892184
- Sjoding MW, Dickson RP, Iwashyna TJ, Gay SE, Valley TS. Racial Bias in Pulse Oximetry Measurement. N Engl J Med. 2020 Dec 17;383(25):2477-2478. doi: 10.1056/NEJMc2029240. Erratum in: N Engl J Med. 2021 Dec 23;385(26):2496. doi: 10.1056/NEJMx210003. PMID: 33326721; PMCID: PMC7808260
Round 2
Reviewer 1 Report
Comments and Suggestions for Authors
Accep
Author Response
Thank you!